# A data driven clinical algorithm for differential diagnosis of pertussis and other respiratory infections in infants

**Alberto Eugenio Tozzi[1], Francesco Gesualdo[1]\*, Caterina Rizzo[1], Emanuela Carloni[1], Luisa Russo[1], Ilaria Campagna[1], Alberto Villani[2], Antonino Reale[3], Carlo Concato[4], Giulia Linardos[4], Elisabetta Pandolfi[1]**

**1** Predictive and Preventive Medicine Research Unit, Multifactorial and Systemic Diseases Research Area, Bambino Gesù Children's Hospital, Rome, Italy, **2** General Pediatrics and Infectious Diseases Unit, Department of Pediatrics, Bambino Gesù Children's Hospital, Rome, Italy, **3** Pediatric Emergency Department, Bambino Gesù Children's Hospital, Rome, Italy, **4** Virology Unit, Laboratory Department, Bambino Gesù Children's Hospital, Rome, Italy

\* francesco.gesualdo@opbg.net

## Abstract

### Background

Clinical criteria for pertussis diagnosis and clinical case definitions for surveillance are based on a cough lasting two or more weeks. As several pertussis cases seek care earlier, a clinical tool independent of cough duration may support earlier recognition. We developed a data-driven algorithm aimed at predicting a laboratory confirmed pertussis.

### Methods

We enrolled children <12 months of age presenting with apnoea, paroxistic cough, whooping, or post-tussive vomiting, irrespective of the duration of cough. Patients underwent a RT-PCR test for pertussis and other viruses. Through a logistic regression model, we identified symptoms associated with laboratory confirmed pertussis. We then developed a predictive decision tree through Quinlan's C4.5 algorithm to predict laboratory confirmed pertussis.

### Results

We enrolled 543 children, of which 160 had a positive RT-PCR for pertussis. A suspicion of pertussis by a physician (aOR 5.44) or a blood count showing leukocytosis and lymphocytosis (aOR 4.48) were highly predictive of lab confirmed pertussis. An algorithm including a suspicion of pertussis by a physician, whooping, cyanosis and absence of fever was accurate (79.9%) and specific (94.0%) and had high positive and negative predictive values (PPV 76.3% NPV 80.7%).

### Conclusions

An algorithm based on clinical symptoms, not including the duration of cough, is accurate and has high predictive values for lab confirmed pertussis. Such a tool may be useful in low

**Data Availability Statement:** The policy of our hospital, in line with the Italian regulations, does not allow data sharing of clinical data when there is a risk of identification of patients even when they

are deidentified. This concern is usually high for rare diseases and decreases with the frequency of diseases. Since, however, identification may depend on other factors as specific patterns of disease in a specific site, requests for data sharing are directly managed by our Data Protection Officer who is in charge of the analysis of requests and relevant decisions. The specific Italian regulations are included in art. 2-septies, comma 8, D Lgs 196/2003 and D Lgs 101/2018. Our Data Protection Officer is Dr Luigi Recupero: dpo@opbg.net

**Funding:** This work was partially supported by the ECDC (European Center for Disease Control) within the Pertinent project (Pertussis in Infants European Network), a European hospital-based network dedicated to measuring pertussis burden in infants and to studying pertussis vaccine effectiveness (framework contract number ECDC/2015/017). The study received additional funding from internal research funds of the Bambino Gesù Children's Hospital. The funders had no role in study design, data collection and analysis, decision to publish, or preparation of the manuscript. There was no additional external funding received for this study.

**Competing interests:** The authors have declared that no competing interests exist.

resource settings where lab confirmation is unavailable, to guide differential diagnosis and clinical decisions. Algorithms may also be useful to improve surveillance for pertussis and anticipating classification of cases.

## Introduction

Pertussis, a highly contagious respiratory infection caused by *Bordetella pertussis*, represents a global public health issue [1,2]. Despite the availability of an efficacious vaccine against *B pertussis*, cases continue to occur widely: in 2014 alone, 24 million cases and 160,000 deaths have been estimated in children younger than 5 years [3]. Since the highest toll in terms of complications and deaths associated with pertussis is exacted on infants, prevention and early diagnosis in this age group are of paramount importance.

Cases of pertussis reported through surveillance systems are widely underestimated, as diagnosis is mainly based on varying levels of clinician's suspicion, reporting practices are different in each country, and access to laboratory tests for confirmation is variable [4–9].

Pertussis is clinically diagnosed through a combination of cough and other symptoms. Studies have shown that a cough duration of 14 or more days is the most sensitive of the clinical characteristics, although cough attributes (e.g. paroxysms, whooping, post-tussive vomiting) may increase specificity of clinical diagnostic criteria [10–13]. In fact, the hallmark of almost all clinical case definitions for surveillance is a cough lasting a minimum of two weeks [14–16].

Pertussis symptoms in infants and young children, frequently leading to hospitalization, overlap with those of viral acute respiratory tract infections, whose differential diagnosis often requires a laboratory confirmation test. Although current evidence suggests to start antimicrobial therapy before laboratory confirmation test results are received if pertussis is suspected in infants [17], the early identification of patients at high risk of pertussis is crucial to appropriately prescribe a lab confirmation test and support clinical decisions about antibiotic therapy which may be unnecessary in cases of a viral infections [18,19].

Since recommendations for diagnosis [9,11,13] rely on a cough lasting two or more weeks in addition to other symptoms, we hypothesized that a data driven algorithm may support the differential diagnosis between pertussis and viral respiratory diseases even when the duration of cough is shorter than 2 weeks. To this aim, we performed a retrospective study in which, based on the review of a case series of pertussis and other respiratory infections, we developed a clinical algorithm to predict a laboratory confirmed pertussis in children < 12 months, and estimated its accuracy.

## Methods

### Study design, setting and population

This is a retrospective cohort study conducted between March 2016 and December 2019 in the Bambino Gesù Children's Hospital, a pediatric research hospital located in Rome, Italy. Italy has a universal, free of charge healthcare system, which includes primary pediatric care for all children. Family pediatricians are available for pediatric visits during office hours. Urgent clinical problems and clinical requests outside office hours or holidays are usually referred to hospitals where ambulatory of emergency room services are offered continuously. Children were enrolled in the study based on surveillance criteria set for a hospital based enhanced

surveillance programme and a wide clinical case definition. We included in the study all children below 12 months of age presenting for outpatient care or at the emergency room with one or more apnoea episodes, or paroxistic cough, whooping, or post-tussive vomiting, irrespective of the duration of cough. Infants with a clinical diagnosis of pertussis by a physician or respiratory symptoms and epidemiological linkage to a confirmed pertussis case were also included. Children of parents not speaking Italian were excluded. Enrolled children underwent a nasopharyngeal aspirate. Samples were collected within 24 hours of hospital admission. Culture for B. pertussis was performed on all aspirates, using Regan-Lowe and Bordet-Gengou selective culture media. For cultures, samples were seeded immediately after collection.

For molecular investigations, samples were processed immediately, or stored at -70°C until performing the test. Nucleic acids were extracted from a 200 µl sample of rhinopharyngeal aspirates and purified, using the EZ1 Virus Mini Kit v. 2.0 on the EZ1 Advanced XL platform (Qiagen, GmbH, Hilden, Germany). Nucleic acid extracts were eluted into 90 µl of buffer and processed immediately.

The presence of *B. pertussis* was investigated using a Bordetella Real Time Polymerase Chain Reaction (RT-PCR) kit, targeting IS481 (Bordetella R-gene™ assay (Argene, Biomerieux, Marcy l'Etoile, France), which tested pertussis together with 16 other respiratory viruses: Respiratory Syncytial Virus (RSV) A and B, influenza virus A and B, human coronavirus OC43, 229E, NL-63 and HUK1, adenovirus, hRV, parainfluenza virus 1-2-3-4, human metapneumovirus-hMPV and human bocavirus-hBoV). The choice of this target is supported by our local epidemiology, where circulation of B. parapertussis or B. holmesii is negligible [6]. After signing an informed consent form, patients' families were interviewed to collect sociodemographic variables (age, sex, gestational age, parent education and employment, patient immunization status against pertussis, date of onset of symptoms, type of feeding at symptom onset, number of households, presence of family members with respiratory symptoms), information on clinical symptoms, recent medical visits, ongoing therapy, and previous pertussis immunization. As most patients visited in the emergency room routinely underwent a full blood cell count, we reviewed the clinical records to extrapolate white blood cell and lymphocyte counts. All children with confirmed pertussis were followed up to monitor the duration of the cough.

The study was approved by the Bambino Gesù Children's Hospital Ethical Committee (protocol n.1064_OPBG_2016).

## Definitions

For the purpose of this study we considered as pertussis cases those with RT-PCR or culture positive for *B pertussis* and at least one respiratory symptom among cough, paroxistic cough, whooping, apnoea, or post-tussive vomiting, irrespective of their duration. We also considered as pertussis cases those having a simultaneous positive RT-PCR test for a respiratory virus.

We defined a viral respiratory disease as the presence of one respiratory symptom among cough, paroxistic cough, whooping, apnoea, or post-tussive vomiting, irrespective of their duration, and a PCR positive for RSV A and B, influenza virus A and B, human coronavirus OC43, 229E, NL-63 and HUK1, adenovirus, hRV, parainfluenza virus 1-2-3-4, human metapneumovirus-hMPV and human bocavirus-hBoV.

To compare our results with surveillance standards, we used three common pertussis case definitions: 1) a diagnosis of pertussis made by a physician; OR cough ≥ 2 weeks AND paroxistic cough, OR whooping OR vomiting [14]; 2) cough ≥ 2 weeks AND paroxistic cough, OR whooping OR vomiting [15]; 3) a diagnosis of pertussis made by a physician; OR apnoea; OR cough ≥ 2 weeks AND paroxistic cough, OR whooping OR vomiting [16].

## Statistical analysis and data mining

We studied the association between clinical symptoms included in pertussis clinical case definitions, duration of cough, fever, the simultaneous presence of leukocytosis and lymphocytosis, and a positive PCR for pertussis. We also included in the analysis other symptoms such as fever, petechiae and subconjunctival hemorrage, and season at the onset of symptoms. We also included in the analysis a dichotomous variable for identifying children with white blood cell count greater than the maximum value for age [20] and a lymphocyte percent greater than 50%.

For this purpose, we applied a logistic regression model in which we included only variables that showed a difference between the two groups with a $p<0.3$ at the univariate analysis. We did not test interactions between variables. We then preprocessed the data set to extract variables to be included in data mining. To this aim, we applied a linear discriminant analysis (LDA) in which we included the same independent variables considered in the logistic regression to calculate the canonical discriminant function. We selected as independent variables to be included in data mining those having an absolute value of correlation $>0.1$ within the discriminant function.

The variables identified through LDA as significantly associated with confirmed pertussis were used to create two decision tree algorithms, one including clinical symptoms only, and a second including white blood cell counts. We used a popular classification algorithm, the Quinlan's C.4.5 algorithm [21], to develop decision trees for classification of pertussis cases based on clinical presentation. To this aim, we used WEKA 3.9 (Waikato Environment for Knowledge Analysis), a comprehensive open source Machine Learning toolkit, written in Java, which includes the J48 algorithm, the Java implementation of Quinlan's C4.5 algorithm [21]. A stratified 10-fold cross-validation was performed to estimate the accuracy of the classification and the confusion matrix, by which we calculated sensitivity and specificity, predictive values, and likelihood ratios. We calculated for comparison the same parameters for clinical case definitions used in surveillance standard. Finally, we calculated the cumulative percentage of cases that satisfied the clinical case definitions at enrolment and after 14 days.

## Results

### Study population

We enrolled a total of 543 patients with respiratory symptoms, of whom 160 (29.5%) had a positive RT-PCR for pertussis. In the remaining 383 patients, 116 (21.4% of the total) had a negative PCR result for both pertussis and viral infections. Among the 267 patients with a viral infection, the following viruses were identified: 195 (35.9% of the total) rhinovirus, 80 (14.7%) respiratory syncytial virus, 39 (7.2%) adenovirus, 30 (5.5%) parainfluenza, 28 (5.1%) metapneumovirus, 17 (3.1%) bocavirus, 14 (2.6%) coronavirus, 13 (2.4%) influenza, and 11 (2.0%) enterovirus.

Cultures were performed on all aspirates, and showed growth of B. pertussis in 54 patients, which corresponds to 33.8% of patients that showed a positive RT-PCR for pertussis. All positive cultures showed a positive RT-PCR for B. pertussis. No *B. holmesii* or *B. parapertussis* were detected in cultures.

The general characteristics of patients and their symptoms by category of infection are shown in Table 1.

We did not observe differences in socio-demographic variables between patients with positive vs negative pertussis RT-PCR. Children with pertussis had more likely received a physician's visit in the previous 7 days. Interestingly, a significant proportion of children with a

**Table 1. General characteristics of children included in the study.**

|  | RT-PCR positive for pertussis | RT-PCR negative for pertussis | P-value |
|---|---|---|---|
| Number of patients | 160 | 383 |  |
| Females, number (percent) | 68 (42.5%) | 187 (48.8%) | 0,178 |
| Graduated mother, number (percent) | 53 (33.8%) | 138 (36.3%) | 0,573 |
| Graduated father, number (percent) | 43 (27.9%) | 100 (26.4%) | 0,717 |
| Age at onset of symptoms, months, median (range) | 2.1 (0–11.0) | 1.8 (0–11.9) | 0,290 |
| Season at onset of symptoms |  |  | < 0.001 |
| Spring | 47 (29.4%) | 125 (32.6%) |  |
| Summer | 58 (36.3%) | 68 (17.8%) |  |
| Autumn | 32 (20.0%) | 95 (24.8%) |  |
| Winter | 23 (14.4%) | 95 (24.8%) |  |
| Cough duration, days, mean (SD) | 12.3 (15.7) | 9.8 (21.5) | < 0.001 |
| Paroxism, number (percent) | 123 (76.9%) | 186 (48.6%) | < 0.001 |
| Whooping, number (percent) | 78 (48.8%) | 57 (14.9%) | < 0.001 |
| Cyanosis, number (percent) | 85 (53.1%) | 98 (25.6%) | < 0.001 |
| Post-tussive vomiting, number (percent) | 68 (42.5%) | 131 (34.2%) | 0.067 |
| Apnoea, number (percent) | 111 (69.4%) | 179 (46.7%) | < 0.001 |
| Subconjunctival hemorrage, number (percent) | 15 (9.4%) | 13 (3.4%) | 0.004 |
| Petechiae, number (percent) | 15 (9.4%) | 15 (4.0%) | 0.012 |
| Fever, number (percent) | 41 (25.6%) | 167 (43.6%) | < 0.001 |
| Pertussis suspected by a physician, number (percent) | 53 (33.1%) | 14 (3,7%) | < 0.001 |
| White cell count > max for age and lymphocites > 50%, number (percent) | 48 (30.8%) | 25 (6.8%) | <0.001 |
| Previous pertussis immunization, number (percent) | 23 (15.0%) | 43 (11.8%) | 0.306 |

negative RT-PCR for pertussis presented with symptoms that are typically associated with pertussis, i.e. paroxistic cough, post-tussive vomiting, cyanosis, whooping or apnoea. Fever was less frequent in pertussis cases, although more than 25% of them had this symptom. Petechiae and subconjuctival hemorrage were observed in a small number of cases and more frequently in pertussis. Pertussis was significantly more frequent in summer months.

The results of the logistic regression model to identify the variables most associated with a confirmed pertussis are illustrated in Table 2. Being visited by another physician who suspected pertussis or blood tests showing leukocytosis and lymphocytosis were highly predictive of a positive RT-PCR test for pertussis. Other typical symptoms such as cyanosis, paroxysm and whooping, and symptom onset during summer months were also significantly associated with confirmed pertussis. Interestingly, duration of cough, post-tussive vomiting, and apnoea were not significantly associated with a confirmed pertussis.

The standardized coefficients of the discriminant function in LDA are shown in Table 3 together with the correlations between each discriminating variable and the standardized canonical discriminant function (structure matrix) which represent the loading of the variable in the discriminant function. In this Table, variables are listed by the absolute size of correlation within function.

We found one statistically significant canonical discriminant function (p<0.001), with a Wilks' lambda = 0.694 and a Cehi-square = 184.25. Through LDA, 78.6% of original patients and 77.1% of cross-validated groups were correctly classified.

The variables significantly associated with a positive PCR for pertussis were used for feeding the Weka J48 algorithm to produce decision trees through the selection of a final set of variables. The results are reported in Fig 1. The algorithm without white blood cell counts (Fig 1A)

**Table 2. Variables associated with positive RT-PCR for pertussis through multivariable logistic regression analysis (aOR; 95% CI).**

|  | aOR | 95%CI | P |
|---|---|---|---|
| Female | 0,55 | 0.34–0.91 | 0.019 |
| Age at onset of symptoms, months | 1.89 | 0.38–9.35 | 0.434 |
| Season at onset of symptoms (ref autumn) |  |  |  |
| Spring | 0.63 | 0.31–1.25 | 0.186 |
| Summer | 2.14 | 1.08–4.25 | 0.029 |
| Winter | 0.45 | 0.21–0.95 | 0.037 |
| Cough duration, days | 1.02 | 0.97–1.08 | 0.388 |
| Paroxism | 2.20 | 1.22–3.98 | 0.009 |
| Whooping | 2.53 | 1.45–4.40 | 0.001 |
| Cyanosis | 2.17 | 1.28–3.69 | 0.004 |
| Post-tussive vomiting | 1.10 | 0.66–1.84 | 0.725 |
| Apnoea | 1.38 | 0.81–2.32 | 0.233 |
| Subconjunctival hemorrhage | 1.21 | 0.43–3.47 | 0.717 |
| Petechiae | 1.44 | 0.49–4.26 | 0.509 |
| Fever | 1.07 | 0.62–1.85 | 0.796 |
| Pertussis suspected by a physician | 5.44 | 2.46–12.03 | <0.001 |
| White cell count > max for age and lymphocytes>50% | 4.48 | 2.19–9.17 | <0.001 |
| Pertussis immunization | 2.64 | 0.93–7.45 | 0.067 |

aOR: adjusted Odds Ratio.

CI: Confidence Interval

identified a suspect of pertussis posed by a physician as a single strong predictive criterion for predicting laboratory confirmation. When the patient presented with no previous suspicion of pertussis, the simultaneous presence of whooping and cyanosis with no fever predicted lab confirmation. This algorithm correctly classified 79.9% of cases and was 94% specific. When adding white blood cell count (Fig 1B), the decision tree for classifying lab confirmed pertussis was more complex with a slightly lower accuracy (79.0%) and a higher specificity (94.6%).

**Table 3. Identification of variables to feed the algorithm through linear discriminant analysis.**

|  | LDA Standardized coefficients | Structure matrix | Variable included in the algorithm |
|---|---|---|---|
| Pertussis suspected by a physician | 0.479 | 0.661 | Yes |
| Whooping | 0.373 | 0.577 | Yes |
| White cell count > max for age and lymphocytes>50% | 0.328 | 0.512 | Yes |
| Cyanosis | 0.244 | 0.42 | Yes |
| Paroxysm | 0.209 | 0.395 | Yes |
| Apnoea | 0.142 | 0.333 | Yes |
| Fever | -0.05 | -0.251 | Yes |
| Post-tussive vomiting | 0.041 | 0.125 | Yes |
| Season at symptom onset (summer vs others) | -0.102 | -0.141 | Yes |
| Cough duration (days) | 0.176 | 0.091 | No |
| Female | -0.172 | -0.08 | No |
| Previous pertussis immunization | 0.101 | 0.014 | No |
| Age at onset of symptoms. Months | 0.509 | 0,003 | No |
| Subconjunctival hemorrhage | 0.035 | 0.181 | No |
| Petechiae | 0.044 | 0.175 | No |

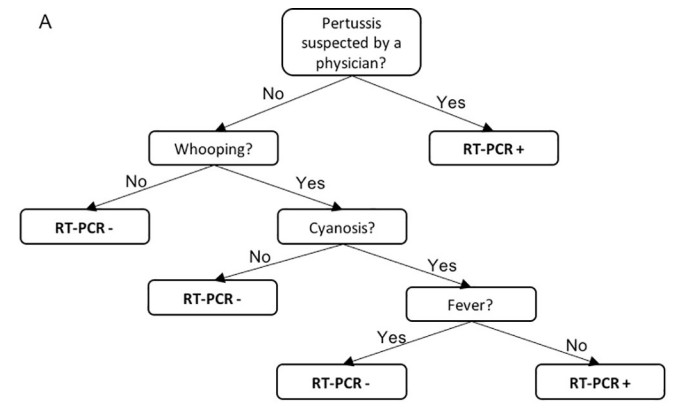

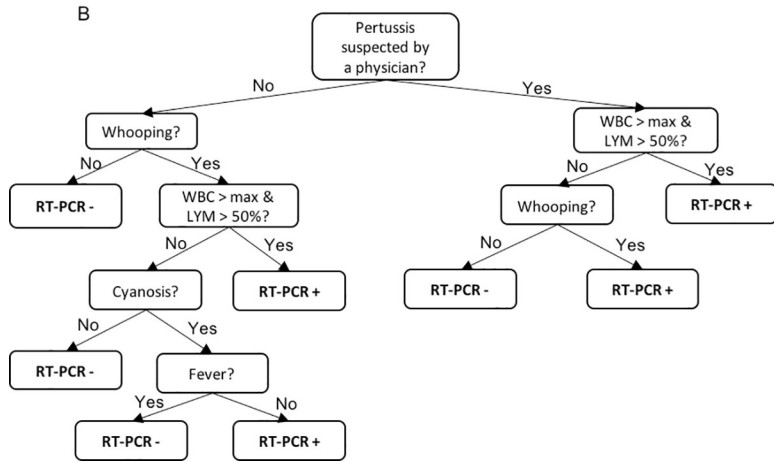

**Fig 1. Clinical algorithm for predicting a case with RT-PCR+ for pertussis.** (A) Clinical algorithm developed with clinical symptoms only. (B) Clinical algorithm developed with the addition of white blood cell count.

We lastly compared the performance of the algorithms with that of the three case definitions considered in the study. Results are reported in Table 4.

**Table 4. Comparison of accuracy of algorithms with some clinical surveillance case definitions for pertussis.**

|  | WHO (14) | CDC (15) | ECDC (16) | Algorithm A | Algorithm B |
|---|---|---|---|---|---|
| Accuracy | 73.30% | 70.53% | 55.25% | 79.93% | 78.39% |
| Sensitivity | 40.63% | 30.63% | 75.62% | 46.30% | 41.70% |
| Specificity | 86.95% | 87.21% | 46.74% | 94.00% | 94.00% |
| PPV | 56.52% | 50% | 37.20% | 76.30% | 74.70% |
| NPV | 77.80% | 75.06% | 82.10% | 80.70% | 79.10% |
| LR+ | 3.11 | 2.39 | 1.42 | 7.72 | 6.95 |
| LR- | 0.68 | 0.795 | 0.52 | 0.571 | 0.62 |

PPV: Positive Predictive Value.

NPV: Negative Predictive Value.

LR+: Positive Likelihood Ratio.

LR-: Negative Likelihood Ratio.

WHO: World Health Organization.

ECDC: European Centre for Disease Prevention and Control.

Compared with our algorithms, the European Centre for Disease Prevention and Control (ECDC) case definition had a lower accuracy and a lower specificity but a higher sensitivity. Both the World Health Organization (WHO) and the Centers for Disease Control and Prevention (CDC) case definitions had lower accuracy, sensitivity and specificity. Moreover, our algorithms had a much higher positive predictive value and positive likelihood ratio than any case definition. Many children with laboratory confirmation for pertussis met a standard case definition only several days after being enrolled in this study. Specifically, 7, 15, and 16 pertussis cases respectively met the ECDC, WHO and CDC clinical case definition only several days after admission, since they mostly presented before 2 weeks of cough duration.

## Discussion

We found that simple algorithms for differential diagnosis of pertussis in infants younger than 1 year of age, developed with data on clinical symptoms and laboratory findings commonly collected in clinical practice, are accurate and specific and have a much higher positive predictive value and positive likelihood ratio than any clinical case definition for surveillance. These algorithms allow for different combinations of symptoms and may be helpful in timely supporting decisions for diagnosis and therapy.

Since a cough duration of at least two weeks is considered a hallmark for identifying pertussis cases, our algorithms may help to earlier identify infants younger than 1 year at high risk of pertussis. Patients are likely to seek health care early in the course of respiratory diseases, and only 33.7% of our pertussis cases had a cough history > 14 days when they were enrolled. Pertussis diagnosis is often missed or delayed, because its clinical picture is similar to other respiratory infections in infancy [18,22]. In our study, typical symptoms for pertussis were also frequently seen in infants with other respiratory conditions. Since pertussis must be ruled out in infants presenting with acute persistent cough [23,24], an algorithmic approach that transforms the generic clinical description of respiratory infections into a standard approach to differential diagnosis may be a powerful clinical tool. Given the high positive predictive values and positive likelihood ratios, the use of these algorithms may be important especially in low resource settings or where laboratory facilities for performing a PCR test or culture are not available. As a matter of fact, our algorithm has been developed in a setting characterised by a high availability of resources, and therefore it can be used in a similar, high resource environment. Nevertheless, the same algorithm can be trained with data originating from different contexts, and can therefore be adapted even to settings with fewer resources.

Our algorithms included symptoms specific for pertussis as a previous suspect of pertussis, whooping, cyanosis, and absence of fever, and leukocytosis and lymphocytosis. When looking at variables associated with pertussis in the logistic regression analysis, a previous suspect of pertussis and leukocytosis plus lymphocytosis were 5 and 4 times more frequent in pertussis cases respectively, while typical symptoms like cyanosis, paroxysm, and whooping were only two times more frequent in cases. Moreover, being a male and being observed in summer were significantly more frequent in lab confirmed pertussis but did not help to predict the presence of the disease. In our setting, apnoea and post-tussive vomiting were not statistically associated with pertussis despite the fact that they are included in most common clinical case definitions. Interestingly, also previous immunization, previous antibiotic treatment, age and duration of cough were not associated with confirmed pertussis. Finally, the absence of fever was helpful in classifying pertussis cases. The majority of infants presenting with respiratory symptoms in the emergency room routinely undergo a blood cell count. It is well known that leukocytosis and lymphocytosis are specific markers of pertussis. The inclusion of this information in the

algorithm, however, increased only slightly its specificity, suggesting that other clinical symptoms are sufficient to support clinical decisions and differential diagnosis.

Our observations may have implications for surveillance. The comparison of our algorithms with other case definitions showed increased accuracy and specificity. Current clinical case definitions for surveillance may miss cases early in the course of the disease and, as a matter of fact, a significant number of pertussis cases fulfilled the clinical case definition for surveillance only several days after the enrollment in this study. Specific algorithms, therefore, may help to improve early diagnosis even in low resource settings [25]. Moreover, the performance of case definitions is affected by the prevalence of the disease. The high positive predictive value and positive likelihood ratio of our algorithms suggest that they may be most useful in high prevalence settings and in specific circumstances as outbreaks, epidemic years or during summer season, when pertussis is most frequent.

In 2011, an algorithm for clinical diagnosis of pertussis was elaborated by the Global Pertussis Initiative (GPI) based on expert opinion to offer diagnostic and surveillance tools tailored to age [26]. A direct comparison between our algorithms and those proposed by GPI is not possible as they differ by symptoms and age groups. On the other hand, our data driven approach, which focused on children below 12 months only, was partially consistent with symptoms in the 4mo - 9yrs age group for the aggregation of whooping, cyanosis and absence of fever.

Ideally, a case definition should have both a high specificity and sensitivity. However, while a high sensitivity is required for a precise estimation of the burden and for studying transmission of disease, a high specificity allows to avoid false positives which do not require intervention. Moreover, a case definition with high specificity is desirable in vaccine evaluations or in studies on disease trends [27].

A possible change of pertussis case definition and duration of cough has long been debated [9,25,28–30]. To our knowledge, an algorithmic approach to refine pertussis case definitions and achieve a better performance has never been used, and deserves attention in perspective.

Our study has several strengths. We worked within an enhanced surveillance framework with standard procedures to detect suspected pertussis cases. We also studied a population in a restricted age range (< 1 year) where pertussis is most severe and differential diagnosis with other viral diseases is most difficult. Pertussis may present with atypical symptoms, particularly in the newborn, when pertussis may be even fatal. Our algorithm even correctly classifies patients in this age group.

Moreover, we simultaneously searched for pertussis and a wide array of viral infections through PCR testing, avoiding misclassification of cases. Finally, we monitored pertussis cases over time to calculate how many of them had satisfied surveillance case definitions after laboratory confirmation.

We used a robust data mining approach to build a decision algorithm, based on routine clinical data. The computational capabilities for data mining have become more accessible as computer power has increased and dedicated software has become available. Compared with a simple list of symptoms, algorithms are more flexible and allow for visualizing complex interactions. The value of these tools may be extremely important when support for clinical diagnosis is associated with evidence based treatment recommendations [31].

On the other hand, this study also has limitations.

First, we did not systematically exclude a diagnosis of B. parapertussis or B. holmesii in the analysed samples. This choice was based on the epidemiology of Bordetella spp. in Italy, where circulation of B. parapertussis or B. holmesii is exceedingly rare. In support of this observation, we recently reviewed a case series of young children with pertussis in our area, using a RT-PCR against IS481 and IS1001, and confirming pertussis positive results with a specific

RT-PCR assay for B. pertussis using the ptxP (promoter of pertussis toxin gene) as target. In this study, we did not find any evidence of infection by B. holmesii or B. parapertussis [6]. The absence of B. holmesii and B. parapertussis was also confirmed by culture in one third of the RT-PCR positive samples. Given all this premises, we believe that, even if a misclassification potential exists, it is negligible.

Moreover, such an algorithm has been developed in a high resource setting and may not universally apply to all populations because of patient selection, access to healthcare facilities, and availability of laboratory confirmation. However, the methods used in this work are transparent and completely reproducible so that a customized algorithm can be developed for any setting, based on the specific setting's data.

As with any case definition using reported symptoms, our algorithm is subject to recall bias and we have not investigated the value of other unusual symptoms. Additional predictive clinical characteristics may be explored through mining large numbers of clinical records in settings where EHR is available.

In conclusion, we have developed a data driven algorithm based on simple clinical information which reliably supports the early differential diagnosis between pertussis and other respiratory conditions in infants younger than 1 year. Such a resource may be important in clinical practice for deciding in a timely manner the most appropriate case management when laboratory confirmation is not available. An algorithmic approach may also be useful for developing novel clinical case definitions for surveillance to partially address their low accuracy.

## Author Contributions

**Conceptualization:** Alberto Eugenio Tozzi, Francesco Gesualdo, Caterina Rizzo.

**Data curation:** Luisa Russo, Ilaria Campagna.

**Formal analysis:** Emanuela Carloni.

**Investigation:** Luisa Russo, Ilaria Campagna, Carlo Concato, Giulia Linardos.

**Methodology:** Caterina Rizzo, Emanuela Carloni.

**Resources:** Alberto Villani, Antonino Reale, Carlo Concato.

**Supervision:** Elisabetta Pandolfi.

**Writing – original draft:** Alberto Eugenio Tozzi, Caterina Rizzo.

**Writing – review & editing:** Francesco Gesualdo, Alberto Villani, Elisabetta Pandolfi.

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
