## [Decision Letter · Decision Letter 0]

7 May 2020

PONE-D-20-05693

A data driven clinical algorithm for differential diagnosis of pertussis and other respiratory infections in infants

PLOS ONE

Dear Dr Francesco Gesualdo,

Thank you for submitting your manuscript to PLOS ONE. After careful consideration, we feel that it has merit but does not fully meet PLOS ONE’s publication criteria as it currently stands. Therefore, we invite you to submit a revised version of the manuscript that addresses the points raised during the review process.

We would appreciate receiving your revised manuscript by June 5. To enhance the reproducibility of your results, we recommend that if applicable you deposit your laboratory protocols in protocols.io, where a protocol can be assigned its own identifier (DOI) such that it can be cited independently in the future. For instructions see: http://journals.plos.org/plosone/s/submission-guidelines#loc-laboratory-protocols

We look forward to receiving your revised manuscript.

Kind regards,

Daniela Flavia Hozbor

Academic Editor

PLOS ONE

2. During our initial internal evaluation of your submission, we noticed a possible error in Abstract (the second sentence of conclusion: 'were lab confirmation is unavailable' should be 'where lab confirmation is unavailable'). Please note that PLOS ONE does not providing copyediting or proofs of accepted manuscripts. We therefore recommend that you carefully review your manuscript and correct any language errors at this time.

4.Thank you for stating the following financial disclosure:

Reviewers' comments:

Reviewer's Responses to Questions

**Comments to the Author**

1. Is the manuscript technically sound, and do the data support the conclusions?

Reviewer #1: Yes

Reviewer #2: Yes

2. Has the statistical analysis been performed appropriately and rigorously? 

Reviewer #1: Yes

Reviewer #2: Yes

3. Have the authors made all data underlying the findings in their manuscript fully available?

Reviewer #1: Yes

Reviewer #2: Yes

4. Is the manuscript presented in an intelligible fashion and written in standard English?

Reviewer #1: Yes

Reviewer #2: Yes

5. Review Comments to the Author

Reviewer #1: The manuscript addresses an interesting topic related to the diagnosis of pertussis. The clinical criteria and the clinical case definition include cough lasting for 14 days or more, but patients usually seek care earlier. What other criteria allow predicting the confirmation of pertussis?

The paper is adequately written although a minor language revision in some points is necessary.

A point that should be discussed more clearly is the specificity of the PCR used. Amplification of an IS481 sequence can be positive in patients infected with Bordetella pertussis but also with B. holmesii. The clinical presentation can be different between both infections and therefore the predictive capacity of each symptom, too. This point deserves a deep discussion.

In the same sense, the possible influence of clinical symptoms caused by B. parapertussis that, given the diagnostic tool used here, have been considered "negative for pertussis" should be analyzed.

When discussing the PCR results in Table 1, reference should be made to B. pertussis / B. holmesii and not to "pertussis"

Reviewer #2: Authors propose an algorithm aimed at predicting a laboratory confirmed pertussis case. Since many case definitions include cough duration of at least 14 days which are usually not archived at the time of physician visit, in most cases patients don´t fulfill case definitions and are misdiagnosed. Moreover this algorithm may be useful in settings where there is no diagnostic PCR available.

Since statistics and informatic model tools is not in my field of expertise, my revision is focused on clinical and laboratory observations, assuming the statistical analysis and data mining is ok.

I understand from the reading that nasopharyngeal samples were stored at -70°C in some cases. But it is affirmed that culture was performed. No results on culture results are shown. Do you have all negative results? I think the storage of samples at a low temperature is the cause of this negative finding.

was there any exclusion criteria employed for patients selection in this study?

During the period of time studied, was there an outbreak year ? could the results showing that the disease is more common in summer be associated with an outbreak ? this observation is interesting since there has been inconclusive information for many years related to seasonality until recent reports showing pertussis rise in spring and summer seasons (Estimating seasonal variation in Australian pertussis notifications from 1991 to 2016: evidence of spring to summer peaks R. N. F. Leong, J. G. Wood, R. M. Turner, and A. T. Newall)

Can the authors confirm there were no co infections of B. pertussis and respiratory virus? the numbers show that, but it is interesting that fever is included in the manuscript as a criteria for pertussis diagnostics since it is a unusual pertussis symptom. Moreover following the proposed algorithm a child with whooping, cyanosis and fever should be classified as pertussis positive Fig 1B. Fever in this algorithm is unnecessary as discriminating variable? please clarify

The socio-demographic variables analysed are not explicited. Did you thought to include them in the algorithm ?

There are some acronyms not specified what they stand for, eg. aOR, ECDC, please clarify

I suggest to add the comparison with surveillance standards defined by experts in GPI (Global pertussis initiative) which includes a different case definition for young children. In reference 9 of your manuscript authors propose an algorithm for diagnosis of pertussis in children from 0-3months and 4 months to 4 years for settings that lack PCR confirmatory assays. Could you discuss similarities and differences with GPI algorithm.

I also suggest to explicit all along the manuscript that results became from analyzing results from children under 1year of age in a high income country meaning that the algorithm could be used in that context.

Lines numbers in the manuscript are not displayed.

6. PLOS authors have the option to publish the peer review history of their article (what does this mean?). If published, this will include your full peer review and any attached files.

Reviewer #1: No

Reviewer #2: No

---

## [Author Response · Author response to Decision Letter 0]

8 Jun 2020

Dear Reviewers,

Thank you very much for your comments. We revised the paper according to your advice, we hope we have fulfilled all of your requests. 

Here follows a point-by-point response to your comments. 

Reviewer #1 

The manuscript addresses an interesting topic related to the diagnosis of pertussis. The clinical criteria and the clinical case definition include cough lasting for 14 days or more, but patients usually seek care earlier. What other criteria allow predicting the confirmation of pertussis? The paper is adequately written although a minor language revision in some points is necessary.

A point that should be discussed more clearly is the specificity of the PCR used. Amplification of an IS481 sequence can be positive in patients infected with Bordetella pertussis but also with B. holmesii. The clinical presentation can be different between both infections and therefore the predictive capacity of each symptom, too. This point deserves a deep discussion. In the same sense, the possible influence of clinical symptoms caused by B. parapertussis that, given the diagnostic tool used here, have been considered "negative for pertussis" should be analyzed. When discussing the PCR results in Table 1, reference should be made to B. pertussis / B. holmesii and not to "pertussis"

We thank the reviewer for the comment, which gives us the opportunity to expand the discussion on this topic. We constantly monitor circulating Bordetella species and we repeatedly reviewed the existing diagnostic procedures during our surveillance program. According to our data, B. holmesii and B. parapertussis seem exceedingly rare in Italy: 

1) We recently reviewed a case series of young children with pertussis in our area and we did not find any evidence of infection by B. holmesii or B. parapertussis (Stefanelli P, Buttinelli G, Vacca P, Tozzi AE, Midulla F, Carsetti R, Fedele G, Villani A, Concato C; Pertussis Study Group. Severe pertussis infection in infants less than 6 months of age: Clinical manifestations and molecular characterization. Hum Vaccin Immunother. 2017;13:1073-1077). In this study, we performed culture + PCR with IS481 and IS1001. To prevent misdiagnosis of B. holmesii as B. pertussis, all samples positive for B. pertussis were confirmed with a specific Real Time PCR assay for B. pertussis using the ptxP (promoter of pertussis toxin gene) as target. 

2) Moreover, all samples collected in our study were cultured (see below) and among the 543 patients included in the study, none had evidence of different isolates than B pertussis. More precisely, 54 cultures showed growth of B. pertussis (out of 154 RT-PCR positive patients) and no B. parapertussis or B. holmesii were detected in cultures.

In light of these comments, we better addressed the diagnostic details through adding: 1) a clarification on cultures and a justification of the choice of the IS481 target in the methods section; 2) details on the results of cultures in the results section; 3) a comment in the discussion (limitation paragraphs) addressing the potential of misclassification of diagnosis. Moreover, we changed the headings in Table 1 to “negative for B. pertussis” and “positive for B. pertussis”.

Reviewer #2

Authors propose an algorithm aimed at predicting a laboratory confirmed pertussis case. Since many case definitions include cough duration of at least 14 days which are usually not archived at the time of physician visit, in most cases patients don´t fulfill case definitions and are misdiagnosed. Moreover this algorithm may be useful in settings where there is no diagnostic PCR available. Since statistics and informatic model tools is not in my field of expertise, my revision is focused on clinical and laboratory observations, assuming the statistical analysis and data mining is ok.

I understand from the reading that nasopharyngeal samples were stored at -70°C in some cases. But it is affirmed that culture was performed. No results on culture results are shown. Do you have all negative results? I think the storage of samples at a low temperature is the cause of this negative finding.

We better clarified how the sampling and the diagnostic procedures were performed. Indeed, all nasopharingeal aspirates were seeded before they were stored at -70° and cultured. The results of cultures have been added in the results section.

was there any exclusion criteria employed for patients selection in this study?

The only exclusion criteria applied was parents not speaking Italian. No other exclusion criteria were applied. We added this information in the text. 

During the period of time studied, was there an outbreak year ? could the results showing that the disease is more common in summer be associated with an outbreak ? this observation is interesting since there has been inconclusive information for many years related to seasonality until recent reports showing pertussis rise in spring and summer seasons (Estimating seasonal variation in Australian pertussis notifications from 1991 to 2016: evidence of spring to summer peaks R. N. F. Leong, J. G. Wood, R. M. Turner, and A. T. Newall)

We agree that this observation is interesting and, although we observe years with higher incidence, pertussis in young children seems still endemic in our area, possibly due to the very low coverage of pertussis immunization in pregnant women. We did observe a higher incidence of B. pertussis in 2016, however the number of cases per year was never negligible and a seasonal trend seems detectable in all years. Even if symptom onset in summer was significantly associated with confirmed pertussis at the logistic regression analysis and at the linear discriminant analysis, this variable did not statistically help to predict pertussis in the algorithms and therefore was not included. We feel that a comment would not really fit the Discussion in its current form, as we mainly discussed the variables that are actually included in the final algorithm. Nevertheless, we are available to add a comment in case the reviewer thinks this is needed. 

Can the authors confirm there were no co infections of B. pertussis and respiratory virus? the numbers show that, but it is interesting that fever is included in the manuscript as a criteria for pertussis diagnostics since it is a unusual pertussis symptom. Moreover following the proposed algorithm a child with whooping, cyanosis and fever should be classified as pertussis positive Fig 1B. Fever in this algorithm is unnecessary as discriminating variable? please clarify

We thank the reviewer for noting a mistake in the figure. A child with whooping, cyanosis, and no fever leads to pertussis in the algorithm. We edited the graphs accordingly.

We did find coinfections and we included them as pertussis as the finding of a B. pertussis positivity always justifies an antibiotic therapy and isolation measures. (This is stated in the Methods section).

In our experience, children with coinfections have a clinical picture which is similar to that of pertussis. More specifically, the proportion of children with fever does not significantly differ between a pertussis only and a co-infection diagnosis (Frassanito A, Nenna R, Nicolai A, Pierangeli A, Tozzi AE, Stefanelli P, Carsetti R, Concato C, Schiavoni I, Midulla F; Pertussis study group. Infants hospitalized for Bordetella pertussis infection commonly have respiratory viral coinfections. BMC Infect Dis. 2017;17:492).

The socio-demographic variables analysed are not explicited. Did you thought to include them in the algorithm ?

As reported in Table 1, the socio-demographic variables collected in this study were age, sex and parent education. We selected the variables for the logistic regression model and for the linear discriminant analysis based on the statistical differences at the univariate analysis, including only those with a p at the statistical test <= 0.30. Only sex and age remained in the models for the other analyses. According to our analyses then, none of these two variables contributed to a better classification of pertussis. We better clarified in the methods how the selection of variables for the algorithms was made and we reviewed Table 3 that now reports the same variables included in the logistic regression analysis..

There are some acronyms not specified what they stand for, eg. aOR, ECDC, please clarify

Accomplished

I suggest to add the comparison with surveillance standards defined by experts in GPI (Global pertussis initiative) which includes a different case definition for young children. In reference 9 of your manuscript authors propose an algorithm for diagnosis of pertussis in children from 0-3months and 4 months to 4 years for settings that lack PCR confirmatory assays. Could you discuss similarities and differences with GPI algorithm.

We added a paragraph on the comparison of this algorithm with ours in the discussion

Overall, a comparison between the GPI’s algorithm and ours is not straightforward, as they differ by symptoms taken into account and age group. GPI also took into account coryza, seizures, pneumonia, coinfection, sleep disturbance and sweating, that we did not record on our patients. Moreover, we developed an algorithm in which cough duration did not help to classify pertussis. GPI algorithm is based on experts’ opinion, while our algorithm is data driven. 

I also suggest to explicit all along the manuscript that results became from analyzing results from children under 1year of age in a high income country meaning that the algorithm could be used in that context.

We thank the reviewer for the suggestion, we stated more explicitly along the manuscript that the algorithm targeted infants younger than on year of age, and that it has been developed in a high resource setting. We also underlined that the same algorithm can be trained with different data, coming from different context, and can therefore be adapted to other settings.

---

## [Decision Letter · Decision Letter 1]

29 Jun 2020

A data driven clinical algorithm for differential diagnosis of pertussis and other respiratory infections in infants

PONE-D-20-05693R1

Dear Dr. Francesco Gesualdo,

We’re pleased to inform you that your manuscript has been judged scientifically suitable for publication and will be formally accepted for publication once it meets all outstanding technical requirements.

Kind regards,

Daniela Flavia Hozbor

Academic Editor

PLOS ONE

Additional Editor Comments (optional):

Reviewers' comments:

Reviewer's Responses to Questions

**Comments to the Author**

1. If the authors have adequately addressed your comments raised in a previous round of review and you feel that this manuscript is now acceptable for publication, you may indicate that here to bypass the “Comments to the Author” section, enter your conflict of interest statement in the “Confidential to Editor” section, and submit your "Accept" recommendation.

Reviewer #1: All comments have been addressed

2. Is the manuscript technically sound, and do the data support the conclusions?

Reviewer #1: Yes

3. Has the statistical analysis been performed appropriately and rigorously? 

Reviewer #1: Yes

4. Have the authors made all data underlying the findings in their manuscript fully available?

Reviewer #1: Yes

5. Is the manuscript presented in an intelligible fashion and written in standard English?

Reviewer #1: Yes

6. Review Comments to the Author

Reviewer #1: (No Response)

7. PLOS authors have the option to publish the peer review history of their article (what does this mean?). If published, this will include your full peer review and any attached files.

Reviewer #1: No

---

## [Editor Report · Acceptance letter]

8 Jul 2020

PONE-D-20-05693R1 

A data driven clinical algorithm for differential diagnosis of pertussis and other respiratory infections in infants 

Dear Dr. Gesualdo:

I'm pleased to inform you that your manuscript has been deemed suitable for publication in PLOS ONE. Congratulations! Your manuscript is now with our production department. 

Kind regards, 

on behalf of

Dr. Daniela Flavia Hozbor 

Academic Editor

PLOS ONE